# Study of 1500V AlGaN/GaN High-Electron-Mobility Transistors Grown on Engineered Substrates

**An-Chen Liu [1], Pei-Tien Chen [1], Chia-Hao Chuang [2], Yan-Chieh Chen [3], Yan-Lin Chen [4], Hsin-Chu Chen [3,\*], Shu-Tong Chang [5], I-Yu Huang [6] and Hao-Chung Kuo [1,7,\*]**

[1] Department of Photonics and Institute of Electro-Optical Engineering, National Yang Ming Chiao Tung University, Hsinchu 30010, Taiwan; arsen.liou@gmail.com (A.-C.L.); dy1998416@gmail.com (P.-T.C.)

[2] Industry Academia Innovation School, National Yang Ming Chiao Tung University, Hsinchu 30010, Taiwan; charleschc.ee06@nctu.edu.tw

[3] Institute of Advanced Semiconductor Packaging and Testing, National Sun Yat-sen University, Kaohsiung 804201, Taiwan; jschen20000415@gmail.com

[4] Master Program in Semiconductor and Green Technology, National Chung Hsing University, Taichung 402202, Taiwan; g111002610@mail.nchu.edu.tw

[5] Department of Electrical Engineering, National Chung Hsing University, Taichung 402202, Taiwan; stchang@dragon.nchu.edu.tw

[6] College of Semiconductor and Advanced Technology Research, National Sun Yat-sen University, Kaohsiung 804201, Taiwan; iyuhuang@mail.nsysu.edu.tw

[7] Semiconductor Research Center, Hon Hai Research Institute, Taipei 114699, Taiwan

\* Correspondence: chenhc@mail.nsysu.edu.tw (H.-C.C.); hckuo0206@nycu.edu.tw (H.-C.K.)

**Abstract:** In this study, we demonstrate breakdown voltage at 1500 V of GaN on a QST power device. The high breakdown voltage and low current collapse performance can be attributed to the higher quality of GaN buffer layers grown on QST substrates. This is primarily due to the matched coefficient of thermal expansion (CTE) with GaN and the enhanced mechanical strength. Based on computer-aided design (TCAD) simulations, the strong electric-field-induced trap-assisted thermionic field emissions (TA-TFEs) in the GaN on QST could be eliminated in the GaN buffer. This demonstration showed the potential of GaN on QST, and promises well-controlled performance and reliability under high-power operation conditions.

**Keywords:** GaN on engineered poly-AlN substrates; QST substrate; GaN on Si substrate; HEMT; high breakdown voltage

## 1. Introduction

Gallium nitride (GaN) has emerged as a crucial material for the next generation of high-frequency and high-power devices due to its exceptional properties, including a high concentration of two-dimensional electron gas (2DEG), superior carrier mobility, low on-resistance, and high breakdown voltage [1–3]. These attributes have propelled GaN to the forefront of semiconductor materials, enabling its utilization across a broad spectrum of high-performance electronic and optoelectronic devices. Traditionally, sapphire and silicon substrates have been favored for GaN device fabrication due to their availability and cost-effectiveness. However, their lower thermal conductivity presents a significant challenge for GaN epitaxy, particularly in achieving thick epi-layer stacks for high breakdown voltages exceeding 1200 V [4].

This challenge has been addressed through the incorporation of carbon-doped GaN (GaN:C) buffer layers with HEMT devices, which primarily improves drain leakage currents and increases the breakdown voltage [5]. To tackle these challenges effectively, enhancing the growth of high-resistivity GaN buffers is crucial for achieving robust

electrical insulation from silicon substrates, characterized by minimal leakage currents and a high breakdown voltage.

In our study, we introduce superlattice buffer layers on Qromis Substrate Technology (QS) substrates to reduce the difference in thermal expansion coefficients. This approach enhances the interfacial quality and electronic properties of GaN-based devices. The superior epitaxial quality of GaN layers on QST substrates, compared to those on Si substrates, is expected to significantly enhance device performance [6,7]. Our analysis revealed the superior performance of AlGaN/GaN high-electron-mobility transistors (HEMTs) with QST substrates, achieving a hard breakdown of up to 1500 V. This improvement can be attributed to the superior thermal expansion coefficient between QST substrates and GaN epitaxy, contributing to the enhanced effectiveness of GaN:C thickness to reduce drain leakage currents compared to Si substrates. It is believed that the improved GaN on QST epitaxy process leads to a higher breakdown value.

## 2. Materials and Methods

The epitaxial layers of the $Al_{0.24}Ga_{0.76}N$/GaN-power high-electron-mobility transistors (HEMTs) were grown on high-thermal-conductivity QST substrates by metal–organic chemical vapor deposition (MOCVD). Prior to the preparation of the buffer and active layers, an AlN nucleation layer (NL) was grown to compensate for lattice mismatch and reduce dislocation density within the fabricated devices. This involved the initial growth of a 60 nm AlN nucleation layer to elevate the conduction band energy within the buffer layer and mitigate leakage currents. The AlN NL served to establish a robust foundation for epitaxial growth, enhancing the electrical isolation and thermal conductivity of the device. Following the nucleation layer, a 2 μm thick AlN/GaN superlattice buffer layer was prepared. Subsequently, buffer layer thicknesses ranging from 1 μm to 3 μm were prepared for a comparison of breakdown voltages. Following this, a 300 nm thick undoped GaN channel layer was grown. The structural diagram is depicted in Figure 1a. Subsequently, the AlGaN barrier layer and GaN cap layer were grown on the channel layer, facilitating the formation of high-electron-mobility and two-dimensional electron gas (2DEG) at the interface with the AlGaN barrier layer. The quality of this layer is crucial for the device's overall performance, particularly in terms of on-resistance ($R_{ON}$).

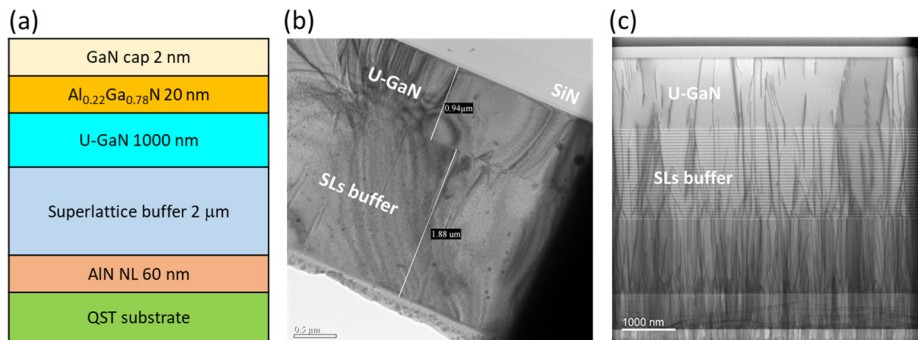

**Figure 1.** The epitaxial structure of (**a**) cross-section schematic view. TEM images of (**b**) GaN on QST and (**c**) GaN on Si substrate.

The composition and thickness of these layers are meticulously calculated to optimize the device's electrical characteristics. The epitaxial layer structure of the GaN on QST and Si substrate and its cross-sectional transmission electron microscopy (TEM) image are illustrated in Figure 1b,c, showcasing the QST substrate high-quality epitaxial growth and the interfaces between the various layers. HEMTs fabricated on QST substrates demonstrate high breakdown voltages and mechanical strength, highlighting the advantages of employing QST for the development of robust and efficient HEMT power devices. This fabrication approach underscores the significance of layer engineering in creating devices

capable of withstanding high voltages and thermal stresses, rendering them suitable for a broad range of high-power applications.

For the fabrication of MIS-HEMTs, the device process commenced with mesa isolation via Ar implantation. Subsequently, source/drain (S/D) ohmic contact formation was carried out using Ti/Al/Ni/Au stacking layers, followed by annealing at 875 °C. After this high-temperature process, ALD-grown 2/20 nm thick $AlN/Al_2O_3$ layers were deposited to serve as both the gate dielectric layer and the first passivation layer [8]. Next, the gate metal was deposited and patterned. Following this, a thick $SiO_2$ inter-layer dielectric (ILD) was applied, followed by the deposition metal 1 (M1), thick $SiO_2$ inter-metal-layer dielectric (IMD), and metal 2 (M2). The M1 layer was 1 mm thick Al and the M2 layer was 2.5 mm thick Al. The backend process followed the CMOS BEOL rule. Finally, a thick $SiN_x$ passivation layer was deposited onto the patterned device structure. The device dimensions, denoted by $L_G/L_{GS}/L_{GD}$, were 2/3.5/22 μm, with these geometric parameters playing a critical role in determining the overall device performance. Figure 2 presents a cross-sectional schematic view of the fabricated devices.

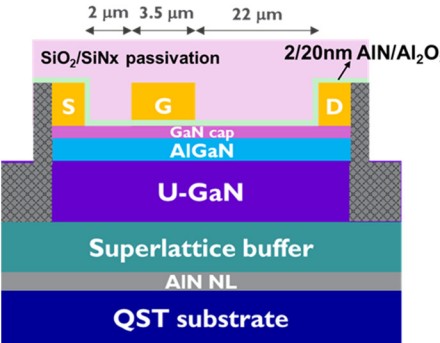

**Figure 2.** Cross-sectional schematic view of the AlGaN/GaN HEMT.

## 3. Result and Discussion

To investigate the impact of QST substrates on device performance, we measured the transfer device characteristics of the fabricated devices, as shown in Figure 3. The threshold voltage was about −12.7 V, and the subthreshold swing was about 98.1 mV/dec, which implies that the device is a well-controlled metal–insulator–semiconductor (MIS) depletion-mode (D-mode) HEMT.

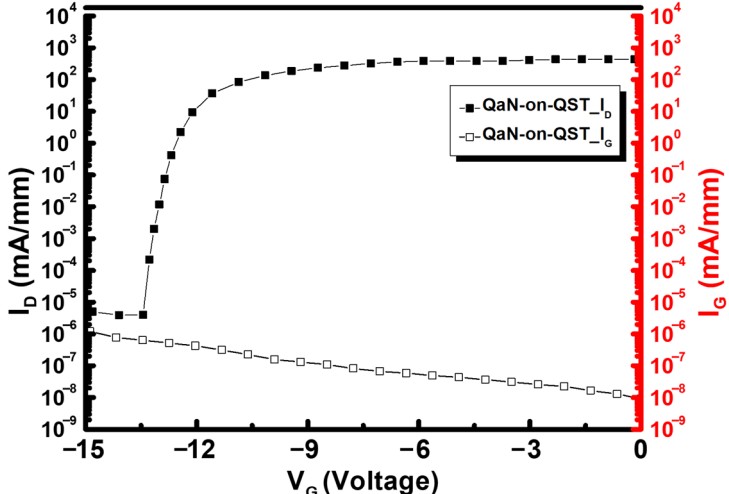

**Figure 3.** Transfer characteristic of AlGaN/GaN HEMT with GaN on QST substrate in semi-log scale: gate leakage current (red line) and drain current (black line).

To demonstrate the superiority of QST substrates over Si substrates, the investigation extended into the off-state behavior of devices. The off-state leakage current was defined as $V_D$ for $I_D$ achieved at 1 µA, as shown in Figure 4a. The variation in GaN:C thickness (1~3 µm) of the $I_{D,off}$ leakage current was analyzed when fixing the SL thickness at 2 µm. It was demonstrated that a thicker GaN:C layer of 3 µm more effectively improved the $I_{D,off}$ leakage current. The obtained values of BV showed a positive correlation to the thickness of GaN:C, which is consistent with a previous report of GaN on Si epitaxy [4]. The breakdown voltages were investigated as a function of the total thickness of the epitaxial structure, as depicted in Figure 4b. The maximum thickness of the GaN buffer layer (SLs+GaN:C) in the QST substrate was 5 µm, whereas the maximum thickness of the GaN buffer layer in the Si substrate was 5.5 µm. A breakdown voltage capability of 1500 V was achieved with GaN on the QST substrate, compared to only 1200 V for GaN on the Si substrate. A linear trend was observed between high breakdown voltage and thickness in GaN on the QST substrate. Therefore, it is believed that a GaN buffer layer thicker than 5 µm on the QST substrate could achieve high performance. A thick GaN buffer layer (>10 µm) will be pursued in future work. Devices grown on highly resistive substrates experience a limited supply of carriers from the depleted region of the substrate, primarily from thermally generated carriers. This leads to substrate depletion [9,10], a phenomenon observed only with highly resistive substrates. As substrate depletion occurs, the electric field on the substrate intensifies, triggering different carrier generation processes such as Shockley–Read–Hall (SRH) generation and/or impact ionization. The QST substrate, composed of materials with a coefficient of thermal expansion (CTE) as a handling layer of the Si(111) layer, exhibits higher resistivity [11]. Consequently, it can withstand a higher electric field, resulting in a higher breakdown voltage than the GaN on Si substrate.

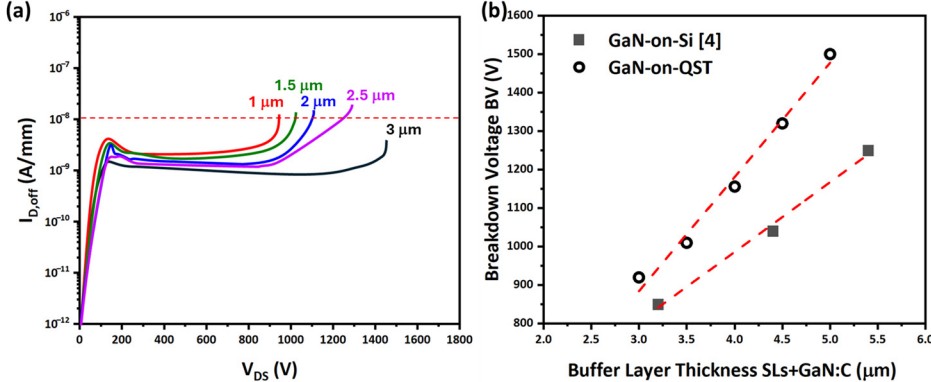

**Figure 4.** (**a**) Off-state $I_D$–$V_D$ characteristics with GaN:C thickness; (**b**) breakdown voltage of epitaxial structures on Si substrate for various epitaxial thickness.

The bowing of epitaxial structures on Si or QST substrates is depicted as a function of the full width at half maximum (FWHM) of X-ray rocking curves from GaN (10–12) diffraction, as shown in Figure 5. Moreover, bowing corresponds to a concave shape; the definition of bowing is illustrated in the inset of this figure. This indicates that threading dislocations alleviate the compressive stress induced in the superlattice (SL) structure and following the thick GaN:C layer during growth, resulting in larger concave bowing with tensile stress due to thermal expansion mismatches during the cooling process after growth. Dislocation dynamics significantly influence the bowing of semiconductor layers. This can be referenced by studies that explore the impact of dislocation density on material bowing and mechanical properties [12,13]. It has been observed that the peak FWHM of the XRD rocking curve remains consistent, while the bowing varies. This indicates that although lattice mismatching is compatible, the resultant thermal coefficient mismatching differs from that observed in GaN on Si structures. Considering QST substrates, it is noted that while the top layer remains Si(111), a thick core layer with a matched CTE is

introduced into the QST substrate. Therefore, the thermal-induced lattice mismatch is mitigated [11]. The wafer bowing of GaN on QST is smaller than that of GaN on Si, despite similar dislocation densities. This is because the CTE layer of the QST substrate enables the release of substrate bowing during the cooling process after the growth of GaN in MOCVD. GaN on QST not only ensures precise stress management in the epitaxial layer structure, but also necessitates improvement in the crystallinity of epitaxial films to grow GaN on Si with reduced bowing. The obtained bowing ranges from 10 to 30 μm for 200 mm wafers, which is sufficiently small for processing in a conventional fabrication line for Si devices.

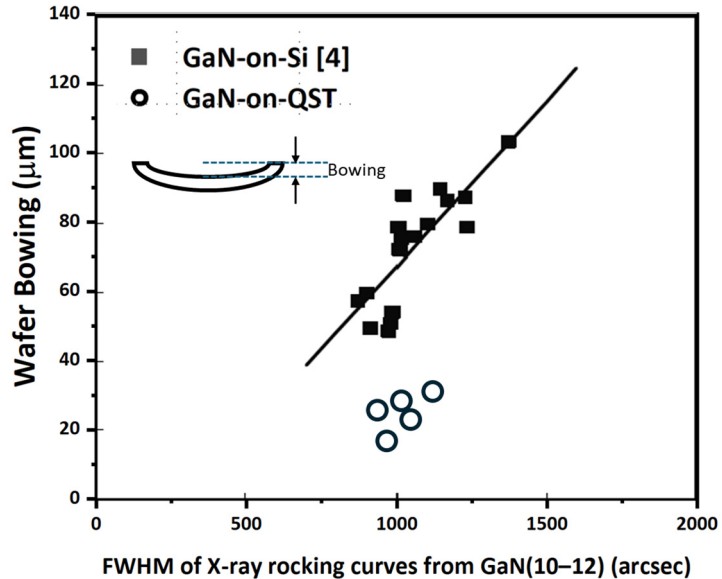

**Figure 5.** Bowing of epitaxial structures as a function of the FWHM of X-ray rocking curves from GaN(10–12).

The pulsed output characteristics obtained at bias points of $(V_{GS}, V_{DS}) = (0\ V, 0\ V)$, $(-12\ V, 0\ V)$, and $(-12\ V, 10\ V)$. The results indicate the presence of trap states that significantly affect the devices' current response to voltage changes. Figure 6 presents the gate and drain lag measurements for GaN HEMTs grown on QST, highlighting their response under varied bias conditions. Specifically, for the GaN on QST sample, the gate lag percentages were 13.3% at $V_D = 6V$ and 10.6% at $V_D = 10V$; drain lag percentages were 31.9% at $V_D = 6V$ and 30.2% at $V_D = 10V$. These findings are crucial for understanding the dynamic behavior of the devices under a range of bias conditions. This behavior suggests that trap-induced lag is a critical factor in device performance, particularly affecting the reliability and operational efficiency under varying electronic loads. The $I_D$ versus the $V_{DS}$ characteristics of GaN HEMTs on QST, when subjected to the aforementioned bias conditions, reveal significant insights, indicating superior buffer layer quality with fewer trap states and a more efficient trap release mechanism. Hence, the advanced buffer layer in the GaN on QST HEMTs correlates with enhanced device performance, characterized by reduced gate/drain lag effects under the specified measurement conditions. This underscores that the buffer layer of GaN on QST substrates providing a more formidable barrier against trap-related degradation, thereby ensuring greater charge carrier mobility and augmented device reliability [14,15].

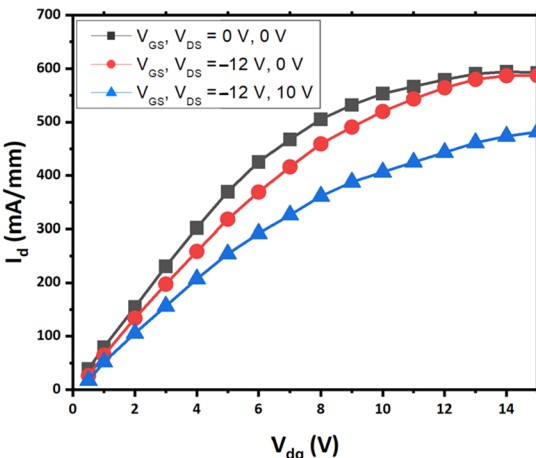

**Figure 6.** Measurement pulsed output characteristics under the bias point: ($V_{GS}$, $V_{DS}$) = (0 V, 0 V), (−12 V, 0 V), (−12 V, 10 V).

QST substrates enhance GaN HEMTs by providing a stable, thermal expansion coefficient platform for growth, reducing defects that cause carrier trapping. Optimized epitaxial growth on QST leads to uniform, high-quality buffer layers, crucial for minimizing trapping and the resulting drain lag.

The low-temperature (5 K) photoluminescence (PL) spectrum served as a diagnostic tool to assess the material quality, carrier concentration, and trap states in GaN HEMT devices. In contrast, we carried out previous work on an optimal GaN on Si device as a reference [16]. Regarding material quality, the sharpness and the positioning of the near-band-edge emission peak were crucial indicators. A notably narrow and intense peak, typically at approximately 360 nm for GaN, denoted a high crystalline quality with minimal defects. For GaN HEMT devices, the spectrum revealed that the PL peak, as shown in Figure 7, for devices on QST substrates was sharper and more pronounced than that on Si substrates, signifying a superior material quality of the QST substrate.

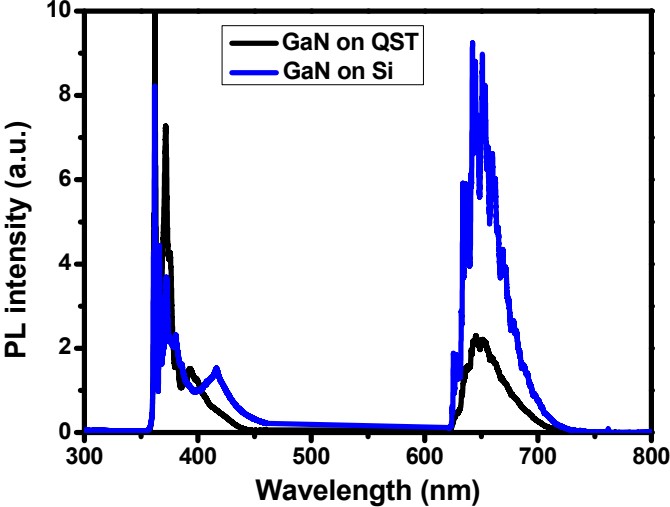

**Figure 7.** PL spectra from the GaN on QST/Si substrate.

As for carrier concentration, the PL peak intensity correlated directly with the radiative recombination rate, which could be linked to the carrier concentration. It was found that higher-intensity peaks were suggestive of elevated carrier concentrations within the material, provided that non-radiative processes were not predominant. Moreover, trap states were inferred from additional peaks in the longer wavelengths, typically in the

visible range, which were indicative of deep-level or defect-related emissions. The PL spectrum for GaN on Si exhibited a broader peak with additional features in comparison to GaN on QST, potentially indicating a higher density of trap states in the Si substrate material. Analyzing the spectral data, the GaN on QST peak was discerned to be narrower and more intense at the band edge, coupled with fewer long-wavelength emissions, thus implying an improved material quality, enhanced carrier concentration, and a reduced number of trap states in comparison to GaN on Si [17,18].

To gain a deeper understanding of the physical phenomenon of TA-TFE in AlGaN/GaN HEMTs of GaN on Si substrate, TCAD simulations were performed under high-bias operation. The simulated device structure of the GaN on Si substrate is the same as that in this study. Simulation models include the drift–diffusion model, polarization, SRH, Auger recombination model, and doping–electric field dependence mobility. The measured breakdown voltage on the device of GaN on Si substrate provides a good calibration of the breakdown dependence TA-TFE model. Y.-H. Li et al. studied the mechanisms of GaN MISHEMT degradation, various negative bias voltages, and various temperatures, as well as dc negative gate bias stress (dc-NGBS) and ac negative gate bias stress (ac-NGBS). The dynamic $R_{ON}$ is higher at higher temperatures, indicating the extracted trap energy levels in the GaN layer due to TA-TFE dominating the degradation of dc-NGBS [19].

The TA-TFE phenomenon occurs due to the increased negative bias applied to the gate. This causes the lateral energy band to elevate beneath the gate region within the GaN layer. Consequently, there is significant bending of the energy band at the channel edge, as illustrated by the lateral energy band depicted in Figure 8c. The cut line referred to is indicated in the x-direction from −4 μm to 14 μm in Figure 8a,b. These findings also contribute to a deeper understanding of semiconductor device physics, particularly emphasizing electron transport mechanisms and leakage paths under strong electric fields [20]. Through TCAD modeling, it was observed that the energy band diagram analysis when $V_G$ was −15 V and $V_D$ was 1200 V showed that the strong electric field caused significant band bending, thereby exacerbating the TA-TFE in the AlGaN/GaN HEMT of the GaN on Si substrate, as shown in Figure 8a. This phenomenon leads to the generation of electron–hole pairs, where electrons are extracted from the drain and holes accumulate in the GaN buffer, especially with the assistance of dislocation traps under high-electric-field conditions. In comparison, in a comparative analysis of AlGaN/GaN HEMT with the GaN on QST substrate, it was found that fewer holes accumulated in the GaN buffer under high electric field conditions, as shown in Figure 8b.

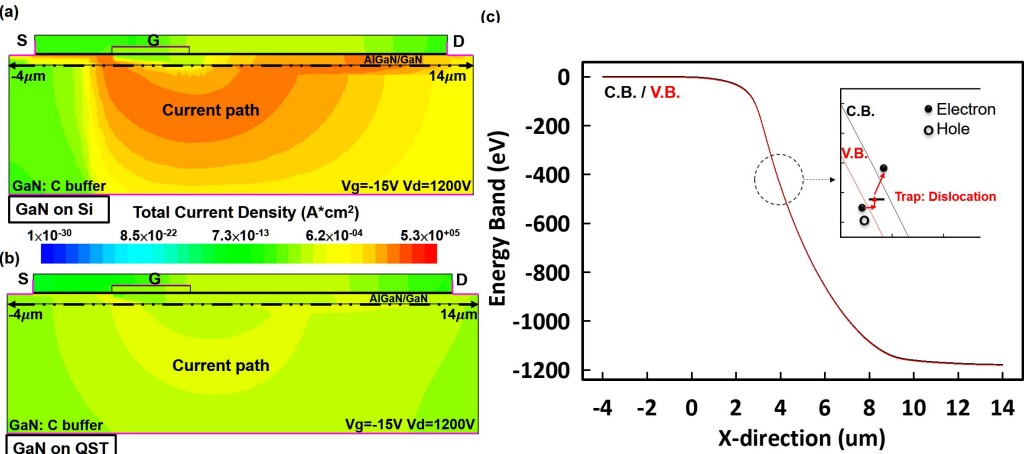

**Figure 8.** The total current density in the GaN buffer of (**a**) the GaN on Si substrate and (**b**) the GaN on QST substrate at $V_G$ = −15 V, $V_D$ = 1200 V. (**c**) Schematic of the energy band across the GaN channel along the horizontal axis and the TA-TFE mechanism.

In order to further study the breakdown mechanism of AlGaN/GaN HEMT on the GaN on Si substrate, especially under the high electric field, the electrical fitting indicated that as the drain voltage increases, the generation of GaN buffer holes also increases directly. This result shows a significant leakage current from the source to the drain. When the level of hole generation in the GaN buffer is low, the increase in leakage current between the source and drain is only slight (shown by the black line), a situation very similar to that of the GaN on QST substrate. These holes are a consequence of recombination, whereby the electrons trapped by acceptor sites are recombined, creating leakage pathways within the buffer, as shown in Figure 9a. Figure 9b displays the fitting parameters as a ratio of h+/[C]. As the ratio approaches 1, the effect of carbon in mitigating the leakage current becomes negligible.

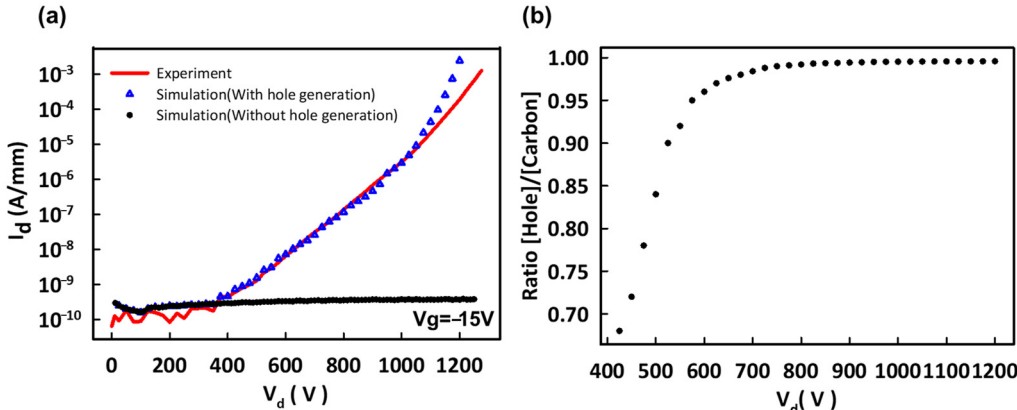

**Figure 9.** (**a**) GaN on Si, GaN on QST (black line) $I_D$-$V_D$ curve, and (**b**) the ratio [hole]/[C] vs. $V_D$ of GaN on Si.

GaN on Si and GaN on QST substrates may exhibit differences in the density and characteristics of traps for electrons and holes caused by dislocations and carbon doping. It has been observed that when the level of hole generation in the GaN buffer is low, the increase in leakage current between the source and drain is insignificant, similar to the behavior observed for GaN on QST substrate, and conversely shown for GaN on Si substrate. However, Figure 9 also demonstrates the efficacy of carbon in reducing the leakage current [20,21]. Therefore, while similarities in trap behavior may exist, differences in the effectiveness of carbon doping in reducing leakage current could contribute to distinct breakdown mechanisms between GaN on Si and GaN on QST substrates under high-electric-field conditions. Due to the superior lattice match and thermal expansion coefficient between QST substrates and GaN epitaxy compared to Si substrates, there may be differences in the effectiveness of carbon doping in reducing leakage current. The improved lattice matching and similar thermal expansion coefficient between QST substrates and GaN may lead to a more structurally intact GaN crystal during epitaxial growth on QST substrates, reducing the formation of dislocations and thereby decreasing the trap density for electrons and holes. In contrast, GaN grown on Si substrates may be more susceptible to dislocation formation, resulting in higher trap densities. Consequently, the effectiveness of carbon doping in reducing leakage current differed due to the distinct characteristics of QST and Si substrates, thereby influencing distinct breakdown mechanisms under high electric field conditions.

## 4. Conclusions

Our research into GaN-based power HEMT devices on QST compared to conventional Si substrates highlights critical insights into device performance, particularly focusing on breakdown voltage capabilities. The GaN on QST devices show a superior high-voltage off-state performance, achieving excellent breakdown voltages of up to VDS =

1500 V at room temperature. Measurements of gate/drain lag and PL have demonstrated the exceptional buffer layer quality of QST substrates. These evaluations revealed a lower prevalence of trap states and a more effective trap release mechanism compared to other substrates. The significant advantage in breakdown voltage for GaN on QST highlights the potential of QST substrates to enhance the performance and reliability of GaN-based power devices. This superior breakdown voltage capability positions GaN on QST as a promising candidate for next-generation high-efficiency power devices, underscoring the importance of substrate technology in advancing semiconductor device performance. Through detailed simulations, we effectively demonstrated how optimized device structures and doping profiles on QST substrates could significantly mitigate substrate leakage and minimize electron injection under high-bias conditions.

**Author Contributions:** Data analysis and structure designing, A.-C.L. and P.-T.C.; fabrication, C.-H.C., Y.-C.C., and Y.-L.C.; writing—review and editing, A.-C.L. and H.-C.C.; supervision, H.-C.K. and S.-T.C.; project administration, H.-C.C., I.-Y.H., and H.-C.K. All authors have read and agreed to the published version of the manuscript.

**Funding:** National Science and Technology Council, Taiwan (112-2218-E-008-007- and 112-2222-E-110-008-).

**Data Availability Statement:** The original contributions presented in the study are included in the article; further inquiries can be directed to the corresponding authors.

**Acknowledgments:** The authors would like to thank Chang Gung University, Taoyuan, along with Hsien-Chin Chiu for the process support; the Taiwan Semiconductor Research Institute (TSRI) for semiconductor-related equipment support; and the Hon Hai Research Institute for their helpful discussion.

**Conflicts of Interest:** The authors declare no conflicts of interest.

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
