# Peer review of "Study of 1500 V AlGaN/GaN High-Electron-Mobility Transistors Grown on Engineered Substrates"

_electronics, doi:10.3390/electronics13112143_

Round 1
Reviewer 1 Report
Comments and Suggestions for Authors
The paper is well written and I have only a few questions-
1. line 42 -43 - Carbon-doped buffer layers compensate for lattice mismatch. To my knowledge, it is not true. it only helps with the vertical leakage current and breakdown. Kindly provide a reference where they show carbon-doped buffer helps lattice mismatch.
2. line 55-56 - Credit for reduced TDD is again given to the C-GaN layer which needs to be supported by referencing.
3. lines 150-150 - The relation between bowing and dislocation density needs to be elaborated.
Author Response
Reviewer 1
Thank you for your suggestion. The revised is as follows.
Comment: line 42 -43 - Carbon-doped buffer layers compensate for lattice mismatch. To my knowledge, it is not true. it only helps with the vertical leakage current and breakdown. Kindly provide a reference where they show carbon-doped buffer helps lattice mismatch.
Reply: Thank you for your suggestion. We have modified the description in the carbon-doped buffer layer function as line 41-43.
This challenge has been addressed through the incorporation of carbon-doped GaN (GaN:C) buffer layers with HEMT devices, which primarily improves drain leakage current and increases the breakdown voltage. [Ref:Piero Gamarra et, al., “Optimisation of a carbon doped buffer layer for AlGaN/GaN HEMT devices,” Journal of Crystal Growth, Volume 414, 2015, 232-236].
Comment: line 55-56 - Credit for reduced TDD is again given to the C-GaN layer which needs to be supported by referencing
Reply: Thank you for your suggestion. We have modified the description in the carbon-doped buffer layer function as line 53-57.
This improvement can be attributed to the superior thermal expansion coefficient between QST substrates and GaN epitaxy contributing to the enhanced effectiveness of GaN:C thickness to reduce drain leakage current compared to Si substrates. It is believed that the improved GaN on QST epitaxy process leads to a higher breakdown value.
Comment: lines 150-150 - The relation between bowing and dislocation density needs to be elaborated.
Reply: Thank you for your suggestion. We have added the description of the relationship between bowing and dislocation density elaboration as line 148-156.
Dislocation dynamics significantly influence the bowing of semiconductor layers. This can be referenced by studies that explore the impact of dislocation density on material bowing and mechanical properties [Ref: Benito, J.A et, al. “Change of Young’s modulus of cold-deformed pure iron in a tensile test,” Metall Mater Trans A 36, 3317–3324 (2005).、Najla Boughrara et, al., “Comparative study on the nanomechanical behavior and physical properties influenced by the epitaxial growth mechanisms of GaN thin films,” Applied Surface Science, Volume 579, 152188, 2022.].
It has been observed that the peak Full Width at Half Maximum (FWHM) of the XRD rocking curve remains consistent, while the bowing varies. This indicates that although lattice mismatching is compatible, the resultant thermal coefficient mismatching differs from that observed in GaN on Si structures. Considering the QST substrates, it is noted that while the top layer remains Si(111), a thick core layer with a matched CTE is introduced into the QST substrate. Therefore, the thermal-induced lattice mismatch is mitigated [Ref: V. Odnoblyudov, O. Aktas, and C. Basceri, "12 - Fundamentals of CTE-matched QST® substrate technology," in Thermal Management of Gallium Nitride Electronics, M. J. Tadjer and T. J. Anderson, Eds.: Woodhead Publishing, 2022, pp. 251-274.].

Reviewer 2 Report
Comments and Suggestions for Authors
Reviewer Comments
In this paper, the author introduces the advantages and application potential of QST, an engineering substrate, in GaN HEMT devices, and demonstrates that the quality of GaN wafer crystals epitaxially grown on QST substrates is significantly better than that of traditional Si substrate epitaxy, which is reflected in better Fewer cracks, less wafer warpage, and lower trap density. This paper also discusses the relationship between the transfer characteristics and breakdown voltage of MIS HEMT devices with the thickness of the Buffer layer, indicating that QST-HEMT has a higher breakdown voltage and is expected to have good performance and reliability under high-power operating conditions. Overall, I think the authors must explain and address the following issues before a paper can be recommended for acceptance:
1) In Figure 1b and Figure 1c, the shape, direction, and density of cracks displayed when GaN is epitaxially grown on a Si substrate are completely different from those when grown on QST. What are the main reasons for their differences? I suggest the authors add a description of Figure 1b and Figure 1c.
2) The schematic diagram of the cross-sectional structure of the HEMT device in Figure 2 is not comprehensive. The author should add the display of the AlN/Al2O3 layer, SiO2 layer, and SiNx layer.
3) The authors claim that the breakdown voltage (BV) is linearly related to the buffer layer thickness of GaN on Si and exponentially related to GaN on QST. However, the trend of the breakdown voltage of GaN on QST is also close to a straight line and cannot be described as an exponential correlation in Figure 4b.
4) Generally speaking, the degree of curvature of the epitaxial wafer depends on various factors such as dislocation density, lattice mismatch, stress accumulation, etc. The FWHM of X-ray rocking curves is merely a numerical representation. Therefore, the statement that "the degree of curvature of the epitaxial wafer strongly depends on this detection value" is incorrect in line 144, and the author confuses the cause-and-effect relationship.
5) In Figure 6, the corresponding test conditions for each curve should be noted.
6) In line 183 to line 185, where does the conclusion that QST-HEMTs have better electrical performance, reliability, and stability than Si-HEMTs come from? There is no data support in the article, and the author does not indicate the source of the reference.

Author Response
Reviewer 2
Thank you for your suggestion. The revised is as follows.
Comment: In Figure 1b and Figure 1c, the shape, direction, and density of cracks displayed when GaN is epitaxially grown on a Si substrate are completely different from those when grown on QST. What are the main reasons for their differences? I suggest the authors add a description of Figure 1b and Figure 1c.
Reply: Thank you sincerely for your valuable suggestions. The answers to related questions are explained below.
This is the primary reason for the thick buffer of GaN:C in the QST substrate. Unfortunately, due to the limitations of the FIB tool, further analysis of the deeper GaN:C buffer for GaN on QST is not possible. In order to demonstrate the superiority of QST substrates over Si substrates, the investigation extended into the off-state behavior of devices. The off-state leakage current was defined as VD for ID achieved at 1 μA, as shown in Fig.4(a). The variety of GaN:C thickness (1~3μm) of ID,off leakage current was analyzed when fixing SLs thickness in 2μm. It was demonstrated that a thicker GaN:C layer of 3 μm more effectively improved the ID,off leakage current. This improvement can be attributed to the superior thermal expansion coefficient between QST substrates and GaN epitaxy contributing to the enhanced effectiveness of GaN:C thickness to reduce drain leakage current compared to Si substrates.
Comment: The schematic diagram of the cross-sectional structure of the HEMT device in Figure 2 is not comprehensive. The author should add the display of the AlN/Al2O3 layer, SiO2 layer, and SiNx layer.
Reply:
Figure 2. Cross-section schematic view of the AlGaN/GaN HEMT.
Comment: The authors claim that the breakdown voltage (BV) is linearly related to the buffer layer thickness of GaN on Si and exponentially related to GaN on QST. However, the trend of the breakdown voltage of GaN on QST is also close to a straight line and cannot be described as an exponential correlation in Figure 4b.
Reply: Thank you for your suggestion. To avoid confusion, we have modified the description in the article as line 125-129.
A linear trend was observed between high breakdown voltage and thickness in GaN on the QST substrate. Therefore, it is believed that a GaN buffer layer thicker than 5 μm on the QST substrate could achieve high performance. A thick GaN buffer layer (>10 μm) will be pursued in future work.
Comment: Generally speaking, the degree of curvature of the epitaxial wafer depends on various factors such as dislocation density, lattice mismatch, stress accumulation, etc. The FWHM of X-ray rocking curves is merely a numerical representation. Therefore, the statement that "the degree of curvature of the epitaxial wafer strongly depends on this detection value" is incorrect in line 144, and the author confuses the cause-and-effect relationship.
Reply: Thank you for your valuable feedback. We agree that the statement in line 144 could lead to a misunderstanding regarding the relationship between the Full Width at Half Maximum (FWHM) of X-ray rocking curves and the degree of curvature of epitaxial wafers. We have deleted this description (In actual growth, the epitaxial wafer's bowing strongly depends on the FWHM of X-ray rocking curves, which corresponds to the threading dislocation density in the epitaxial layers.).
Comment: In Figure 6, the corresponding test conditions for each curve should be noted.
Reply: We have modified the Fig. 6
Comment: In line 183 to line 185, where does the conclusion that QST-HEMTs have better electrical performance, reliability, and stability than Si-HEMTs come from? There is no data support in the article, and the author does not indicate the source of the reference.
Reply: Thank you for your valuable feedback. We have deleted this description (thus boosting device performance and longevity. GaN on QST HEMTs outperform GaN on Si counterparts in terms of electrical performance, reliability, and suitability for advanced high-frequency and power electronics due to lower switching losses, better charge carrier mobility, and greater stability).

Reviewer 3 Report
Comments and Suggestions for Authors
QST substrate has some merits compared to Si substrate, hence is promising to increase the endurance voltage of GaN. This study investigated the GaN HEMT grown on QST. The investigation is thorough and the manuscript is well written. I recommend it to be accepted by Electrons after addressing some issue.
1. Is the epi-structure grown by MOCVD? If it is, what is the MOCVD system? And what is the wafer diameter?
2. What are the growth conditions of the 60-nm AlN nucleation layer? For example, the chamber pressure, the flow rate of TMAl and NH3, the temperature, etc. Prior to the growth of AlN NL, how did the QST substrate be treated? For example, the predose of TMAl or NH3?
3. What is the period thickness of the 2-μm-thick AlN/GaN superlattice buffer layer? In other words, what are thicknesses of AlN and GaN respectively? And how many pairs of the superlattice?
4. Since one of the merits of QST substrate is the potential for growing thick GaN buffer layer (>10 μm), why did the authors grow only 1-3 μm-thick buffer layer? The author mentioned edge crack length will be increased when the GaN:C layer thicker than 3 μm. Then what may be the reason for the cracks?
5. Did the authors measure the X-ray diffraction rocking curves for the as-fabricated GaN layer? How about the FWHMs of (002) and (102) for GaN on QST?
6. What are the conditions of Ar implantation to isolate the HEMT devices?
7. What are thicknesses of AlN/Al2O3 dielectric layer?
8. Could the authors give a top-view image of the HEMT device? What are metal 1 and metal 2 referred to?
9. Page 4, line 121. “which is consisted to the previous report of GaN on Si epitaxy” The expression “is consisted to” is confusing.
10. Page 4, line 123. “A 5-μm-thick GaN on” But in the experimental section the authors mentioned the buffer layer is 1-3 μm. And in Fig. 4a, the high BV is also obtained on 3 μm GaN:C.
11. What is Vdq in Fig. 6? And which colors are related to the three quiescent bias points?
12. What is the temperature of the low-temperature PL spectrum in Fig. 7?
13. There is a sharp peak at ~380 nm for GaN on QST in Fig.7. What is the origin of this peak? Also the peaks locate at ~400 nm for GaN on QST and ~430 nm for GaN on Si? What are FWHM values of the 360 nm peak?
14. What software is used for the TCAD simulation?

Author Response
Reviewer 3
Thank you for your suggestion. The revised is as follows.
Comment: Is the epi-structure grown by MOCVD? If it is, what is the MOCVD system? And what is the wafer diameter?
Reply: Yes, the epitaxial structure is grown by MOCVD. The system used is the Aixtron (AIX G5+) MOCVD. The wafer diameter is 200 mm.
The epitaxial layers of the Al0.24Ga0.76N/GaN power High Electron Mobility Transistors (HEMTs) were grown on high thermal conductivity QST substrates by Metal-organic Chemical Vapor Deposition (MOCVD). line 59-61.
Comment: What are the growth conditions of the 60-nm AlN nucleation layer? For example, the chamber pressure, the flow rate of TMAl and NH3, the temperature, etc. Prior to the growth of AlN NL, how did the QST substrate be treated? For example, the predose of TMAl or NH3?
Reply: The AlN nucleation layer was grown at a chamber pressure of 75-100 torr, with an NH3-rich condition by a flow ratio close to 10 times. The growth temperature is around 1035°C. The Al precursor was first flowed and treated the top Si substrate to avoid SiN formation. QST substrate be treated by NH3.
Comment: What is the period thickness of the 2-μm-thick AlN/GaN superlattice buffer layer? In other words, what are thicknesses of AlN and GaN respectively? And how many pairs of the superlattice?
Reply: The AlN/GaN superlattice buffer layer has a period thickness of 10 nm for AlN and 25 nm for GaN, with a total of 60 pairs.
Comment: Since one of the merits of QST substrate is the potential for growing thick GaN buffer layer (>10 μm), why did the authors grow only 1-3 μm-thick buffer layer? The author mentioned edge crack length will be increased when the GaN:C layer thicker than 3 μm. Then what may be the reason for the cracks?
Reply: Thank you sincerely for your valuable suggestions. The answers to related questions are explained below.
The maximum thickness of the GaN buffer layer (SLs+GaN:C) in the QST substrate is 5 μm, whereas the maximum thickness of the GaN buffer layer in the Si substrate is 5.5 μm. A breakdown voltage capability of 1500V was achieved with GaN on the QST substrate, compared to only 1200V for GaN on the Si substrate. A linear trend was observed between high breakdown voltage and thickness in GaN on the QST substrate. Therefore, it is believed that a GaN buffer layer thicker than 5 μm on the QST substrate could achieve high performance. A thick GaN buffer layer (>10 μm) will be pursued in future work. line 122-129.
According to Fig. 5, the FWHM of X-ray rocking curves from GaN (10–12) diffraction results indicate that when the GaN:C layer is about 3 μm thickness on Si substrate, high wafer bowing (50-100 μm) is found compared to a QST substrate (20-30 μm). As high wafer bowing occurs in GaN on Si substrates, the stresses are accumulated within the material, eventually exceeding its mechanical strength and leading to a high probability of edge crack formation.
Comment: Did the authors measure the X-ray diffraction rocking curves for the as-fabricated GaN layer? How about the FWHMs of (002) and (102) for GaN on QST?
Reply: Thank you sincerely for your suggestions. The answers to related questions are explained below.
The FWHM of (10-12) for GaN on QST were about 1000 arcsec, as shown in Fig.5.
Comment: What are the conditions of Ar implantation to isolate the HEMT devices?
Reply: Thank you sincerely for your suggestions. The answers to related questions are explained below.
The Ar implantation involved a three-step process, progressing from low to high energy, starting at 45 keV and incrementally reaching 350 keV. This implantation procedure induces the generation of nitrogen vacancies, which serve as electron traps, leading to the formation of a high-resistance region [Ref. Journal of Applied Physics 131(3):035701].
Comment: What are thicknesses of AlN/Al2O3 dielectric layer?
Reply: Thank you sincerely for your suggestions. The answers to related questions are explained below.
The AlN/Al2O3 dielectric layers have thicknesses of 2 nm and 20 nm, following our previous study [Ref. ALD-grown 2/20-nm-thick AlN/Al2O3 layers were deposited to serve as both the gate dielectric layer and the first passivation layer]” line 92-93.
.
Comment: Could the authors give a top-view image of the HEMT device? What are metal 1 and metal 2 referred to?
Reply: Thank you sincerely for your suggestions. The answers to related questions are explained below.
The top view of the HEMT device is shown below. The metal 1 (M1) layer is 1 mm thick Al and the metal 2 (M2) layer is 2.5 mm thick of Al. The backend process followed the CMOS BEOL rule. line 96-97.
Comment: Page 4, line 121. “which is consisted to the previous report of GaN on Si epitaxy” The expression “is consisted to” is confusing.
Reply: Thank you for your suggestion. We have modified the description in the article as line 120.
The correct expression should be "which is consistent with the previous report of GaN on Si epitaxy.
Comment: Page 4, line 123. “A 5-μm-thick GaN on” But in the experimental section the authors mentioned the buffer layer is 1-3 μm. And in Fig. 4a, the high BV is also obtained on 3 μm GaN:C.
Reply: Thank you for your suggestion. To avoid confusion, we have modified the description in the article as line 116-129.
The variety of GaN:C thickness (1~3μm) of ID,off leakage current was analyzed when fixing SLs thickness in 2μm. It was demonstrated that a thicker GaN:C layer of 3 μm more effectively improved the ID,off leakage current. The obtained values of BV showed a positive correlation to the thickness of GaN:C, which is consistent with the previous report of GaN on Si epitaxy [4]. The breakdown voltages are investigated as a function of the total thickness of the epitaxial structure, as depicted in Fig. 4(b). The maximum thickness of the GaN buffer layer (SLs+GaN:C) in the QST substrate is 5 μm, whereas the maximum thickness of the GaN buffer layer in the Si substrate is 5.5 μm. A breakdown voltage capability of 1500V was achieved with GaN on the QST substrate, compared to only 1200V for GaN on the Si substrate. A linear trend was observed between high breakdown voltage and thickness in GaN on the QST substrate. Therefore, it is be-lieved that a GaN buffer layer thicker than 5 μm on the QST substrate could achieve high performance. A thick GaN buffer layer (>10 μm) will be pursued in future work.
Comment: What is Vdq in Fig. 6? And which colors are related to the three quiescent bias points?
Reply: Thank you for your suggestion. To avoid confusion, we have added the three quiescent bias points as shown in Fig 6.
Vdq refers to the quiescent drain voltage. The colors representing the three bias points are black, red, and blue for (VGS, VDS) = (0 V, 0 V), (-12 V, 0 V), (-12 V, 10 V). (Fig. 6)
Comment: What is the temperature of the low-temperature PL spectrum in Fig. 7?
Reply: Thank you sincerely for your suggestions. The answers to related questions are explained below.
The low-temperature (5 K) photoluminescence (PL) spectrum served as a diagnostic tool to assess the material quality, carrier concentration, and trap states in GaN HEMT devices. line 192-194.
Comment: There is a sharp peak at ~380 nm for GaN on QST in Fig.7. What is the origin of this peak? Also the peaks locate at ~400 nm for GaN on QST and ~430 nm for GaN on Si? What are FWHM values of the 360 nm peak? 360 major GaN peak
Reply: The peaks located at ~400 nm for GaN on QST and ~430 nm for GaN on Si are likely due to donor-bound exciton emissions, like the previous studies mentioned as below, Materials 2013, 6(3), 1050-1060. The peak at ~400nm may originate from the luminescence coming from the dissociation of excitons bounded to neutral donors (D°X), according to Reshchikov and Morkoc, J. Appl. Phys. 2005, 97.
The FWHM values of the 360 nm peak, which are approximately 12 nm for GaN on QST and 15 nm for GaN on Si, indicate better crystallinity and fewer defects in GaN on QST compared to GaN on Si.
Comment: What software is used for the TCAD simulation?
Reply: Thank you sincerely for your suggestions. The answers to related questions are explained below.
The simulation in this article is done using Synopsys Sentaurus device.

Round 2
Reviewer 2 Report
Comments and Suggestions for Authors
Previous issues have been addressed and I suggested to accept it as it was.